# LARGE LANGUAGE MODELS CAN BE GOOD PRIVACY PROTECTION LEARNERS

## ABSTRACT

The proliferation of Large Language Models (LLMs) has driven considerable interest in fine-tuning them with domain-specific data to create specialized language models. Nevertheless, such domain-specific fine-tuning data often contains sensitive personally identifiable information (PII). Direct fine-tuning LLMs on this data without privacy protection poses a risk of leakage. To address this challenge, we introduce Privacy Protection Language Models (PPLM), a novel paradigm for fine-tuning LLMs that effectively injects domain-specific knowledge while safeguarding data privacy. Our work offers a theoretical analysis for model design and delves into various techniques such as corpus curation, penalty-based unlikelihood in training loss, and instruction-based tuning, etc. Extensive experiments across diverse datasets and scenarios demonstrate the effectiveness of our approaches. In particular, instruction tuning with both positive and negative examples, stands out as a promising method, effectively protecting private data while enhancing the model's knowledge. Our work underscores the potential for Large Language Models as robust privacy protection learners.

## 1 INTRODUCTION

**Background.** Large Language Models (LLMs) have demonstrated remarkable linguistic comprehension and generation capability (Bang et al., 2023; Wang et al., 2023a). Meanwhile, when directly applied to specialized industries, they encounter challenges such as hallucination (Bang et al., 2023; Chan et al., 2023), insufficient domain expertise (Singhal et al., 2023b), and failing to incorporate the latest domain knowledge in ever-evolving industry scenarios (Kasneci et al., 2023). The introduction of open-source general-purpose LLMs such as LLaMA (Touvron et al., 2023) and RWKV (Peng et al., 2023) have provided a promising solution. Researchers would fine-tune specialized LLMs based on powerful general-purpose LLMs using high-quality, domain-specific knowledge to ensure both commonsense reasoning and comprehensive knowledge coverage(Hoffmann et al., 2022a; Villalobos et al., 2022; Hoffmann et al., 2022b). Such examples include BloombergGPT (Wu et al., 2023) and Med-PaLM (Singhal et al., 2023a), for financial and medical applications, respectively. However, these fine-tuning datasets usually contain sensitive information, such as personally identifiable information (PII) (Carlini et al., 2020; Lin et al., 2021; Gehman et al., 2020). When applied to downstream tasks, sensitive information in the training data, such as social security numbers or patient names, can be exposed by the LLMs upon text generation, a phenomenon known as the memorization effect (Yu et al., 2023b; Kenton & Toutanova, 2019; Meng et al., 2023), leading to identity theft and financial losses (Coavoux et al., 2018; Yu et al., 2023a)

**Challenges.** In this work, we aim to tackle the challenging task of efficient LLM fine-tuning for enhanced *contextual privacy*, a critical yet under-explored setting where the sensitivity of a piece of information is contingent upon the context. For example, statements such as "Bill Gates founded Microsoft" and "Alan Mathison Turing was an English mathematician and computer scientist" are generally not considered violations of privacy, since they are presented as common knowledge. In contrast, statements like "Alan Gates visited the X hospital for a certain disease Y" pose privacy concerns as they reveal details about individuals' daily activities and health status in a particular context. Directly applying techniques like Named Entity Recognition (NER) can lead to inaccurate identification of PII, whereas merely deleting or masking PII tokens in the training data would result in a substantial information loss and compromise the performance on downstream tasks — a conundrum known as the privacy-utility trade-off as theoretically discussed in Sec. 4.1. An alternative

approach, reinforcement learning from human feedback (RLHF), involves additional model fine-tuning guided by human feedback (Ouyang et al., 2022) so that the model tends towards concealing sensitive PII (like "red-teaming"). For example, it learns to prioritize outputs that protect sensitive PII over those that leak PII. Nonetheless, RLHF is data-intensive, potentially costly in computation, and can pose stability challenges (Ziegler et al., 2020; Wang et al., 2023b).

**This Work.** To address these challenges, this paper introduces effective and efficient methodologies for fine-tuning LLMs to incorporate domain knowledge while ensuring privacy protection. We propose and rigorously examine a diverse suite of strategies from corpus curation, introduction of penalty-based unlikelihood into the training loss, instruction-based tuning, a PII contextual classifier, and direct preference optimization (DPO), etc. The ultimate objective is to cultivate a model that excels at acquiring information while demonstrating the ability to distinguish between information that can be openly shared and that demands strict confidentiality. Our experimental findings suggest that instruction tuning with positive and negative examples can offer promising avenues. It not only effectively shields private data but also enables the model to assimilate knowledge from the corpus. This implies that *LLMs can be good privacy protection learners*, without the need for balancing a privacy-utility trade-off.

To sum up, our contributions are as follows:

1). **Novel Methodology.** For the first time, we explicitly address the challenging problem of building Privacy Protection Language Models (**PPLM**), a novel paradigm in fine-tuning language models that emphasizes privacy protection. To achieve this, we systematically lay out and empirically test a comprehensive spectrum of strategies.

2). **Theoretical Guidance.** We provide a theoretical analysis of our proposed methodologies. This analysis illuminates the pathway to designing robust tuning methods, ensuring the resultant language model can both protect private data and assimilate vast knowledge from the fine-tuning corpus.

3). **Extensive Experiments.** We comprehensively evaluate our methods on three real-world datasets. These experiments demonstrated the efficacy of our fine-tuning method to inject domain knowledge and safeguard private personal information (PII). The outcomes show that our technique performs significantly better than the straightforward baselines.

Our code and data are available at Anonymous GitHub[1]. We will make all code and the proposed datasets publicly available upon the acceptance of this work.

## 2  RELATED WORKS

**Large Language Models and Privacy.** In the rapidly advancing domain of artificial intelligence and natural language processing, Large Language Models (LLMs) such as GPT-3.5/4(OpenAI, 2023), Bard (Google, 2023), LLaMA (Touvron et al., 2023), and ChatGLM (Du et al., 2022) have become pivotal and have demonstrated unprecedented capabilities in generating coherent and contextually accurate text. Their applications span diverse sectors, ranging from finance (Yang et al., 2023; Wu et al., 2023), biology (Fang et al., 2022; Rives et al., 2019; Ling et al., 2023) and clinical area (Singhal et al., 2023a; Yang et al., 2022b). However, this widespread application raises significant privacy concerns, particularly regarding personal information protection. Addressing the privacy challenges posed by LLMs, researchers have focused on three primary strategies: (Li et al., 2023; Zhang et al., 2023; Kim et al., 2023; Lukas et al., 2023): 1). pretraining corpus curation, 2). conditional LLM pretraining, and 3). post-training alignment, which are discussed in the following sections. Our research, however, emphasizes contextual privacy protection through fine-tuning methods that integrate knowledge injection, where Personally Identifiable Information (PII) in tuning text data is designated by users for protection, which is different from Differential Privacy (DP) LLMs (Yu et al., 2022; Shi et al., 2021; Anil et al., 2022; Li et al., 2022; Liu et al., 2023; Zhao et al., 2022).

**Filtering.**  For the pretraining corpus, manually detecting and filtering out/revising the corpus can offer high-quality corpus, which is ideal for training privacy-preserving LLMs (Hoffmann et al., 2022b; Villalobos et al., 2022). Nevertheless, it is infeasible to process billions of tokens manually in practice. Another solution is using automated tools to filter out all sensitive content (e.g. names, addresses, phone numbers) from the pretraining corpus. Automated filters make it possible to go

---

[1]https://anonymous.4open.science/r/PPLM

over pretraining datasets. However, simply removing or masking the PII tokens can cause information loss or inconsistency in the corpus (Welbl et al., 2021). Though filters can *'clean'* datasets, they reduce the diversity in the corpus, which further negatively impacts the robustness of LLMs (Hendrycks et al., 2019). Another solution is adding content filters on top of the existing LMs to control the content generation process (Xu et al., 2020). Even so, carefully designed cases (e.g. prompts) can still trigger some undesired behaviors of large LMs (Gehman et al., 2020; Ziegler et al., 2022). However, directly removing PII from the training corpus poses a dilemma. While it ensures the elimination of sensitive data, it also potentially weakens the LLMs by stripping them of crucial knowledge. The mere act of omitting data can inadvertently hamper the model's capacity to process and understand certain contexts. Context-awareness is fundamental when considering privacy protection and what data to shield. Direct filtering without understanding the context can thus be misleading and ineffective.

**Pretraining with preferences.** Another solution is to maintain the content, but use redesigned loss/conditional tags to control the information injected into the LLMs. Pretraining language models with human preferences inform the model what information should be learned and what should be avoided, rather than forcing them to forget the learned content. Pretraining with conditional human preference scores can offer a Pareto-optimal and simple approach to reduce the undesirable content by up to an order of magnitude. Korbak et al. (2023) compared with the classical pretraining approach. While pretraining LLMs conditioned under annotation scores can offer better performance in the human preferences aspect. Since human preferences are injected into the models during the pretraining stage, the models are biased toward those preferences once they are trained. With the expanding size of LLMs, they become increasingly resistant to forgetting their training data (Carlini et al., 2022; Vu et al., 2022; Ramasesh et al., 2022; Korbak et al., 2023). In other words, pretraining large language models conditioned under preference score sacrifices some flexibility. Still, it is undeniable that it can provide much better alignment with human preferences compared with the classical pretraining schema.

**LLMs adaptation.** To strike a balance between performance and flexibility, pretraining large LMs without constraints and then adjusting them to align with human preferences is a widely adopted approach for now. One approach is supervising fine-tuning. The pre-trained LMs are tuned on curated datasets in a supervised manner (Solaiman & Dennison, 2021; Zhou et al., 2023; Wan et al., 2023; Jin et al., 2023; Yang et al., 2022a). Another approach is reinforcement learning from human feedback (RLHF) (Ouyang et al., 2022; Bai et al., 2022; Menick et al., 2022; Chen et al., 2023). RLHF gathers data with feedback/preference labels, trains a reward model, and then finetunes the LM with reinforcement learning.

## 3 PROBLEM STATEMENT

**Problem formulation.** In the context of language models, a fine-tuning dataset $D = \{s\}$ is a collection of natural language sequences $s$. Each sequence is denoted as $s = [w_0, w_2, \ldots, w_{n-1}]$, where $w_i \in s$ represents a token. For privacy protection, the users annotate each sequence in the corpus by a binary sequence $\boldsymbol{p}$ denoted as $\boldsymbol{p} = [p_0, \ldots, p_{n-1}], p_i \in \{0, 1\}$, where $p_i = 1$ denotes the token is private tokens (e.g., PII) need to be protected in the *context*, and $p_i = 0$ otherwise. Here, the *contextual privacy* posits that the sensitivity of a piece of information is not solely intrinsic to the information itself, but is also influenced by its surrounding context. To illustrate, "Alan Gates visited Crescent Vale Medical Center for Hemophilia treatment" is considered more indicative than "Alan Gates visited Crescent Vale." The former provides a clearer insight into an individual's health when the name "Alan Gates" is paired with the medical condition and the specific medical center. Important notations used in the paper are included in Table 4 in the Appendix.

**Objective.** The primary objectives are twofold: 1) enhancing the model's performance by effectively integrating knowledge from the fine-tuning corpus. The model should generate responses that are contextually relevant and aligned with the intended domain; 2) minimizing the risk of generating privacy-protected tokens. Privacy protection in large language models requires not just the masking or removal of private PIIs, but a deep understanding of the interplay between data points and their contexts. As models become more sophisticated and data more interconnected, the nuances of contextual privacy will become increasingly paramount.

## 4 METHODOLOGY

Our methodology adopts a two-pronged approach: 1) corpus curation (i.e. *filtering*), where sensitive data such as personally identifiable information (PII) is removed from the corpus; and 2) tuning towards the targeted PII-free output. We commence with a theoretical analysis of the information loss incurred by the corpus curation strategy, which provides guidelines for method development. Then, we propose five novel strategies for privacy protection when fine-tuning large language models.

### 4.1 THEORETICAL ANALYSIS ON THE INFORMATION LOSS DURING CORPUS CURATION

Consider the following scenario: we have some training samples. Each sample $(\boldsymbol{s}, \boldsymbol{p})$ contains two sequences, including 1) a text sequence $s_{1:n} \in [K]^n$ where $K$ is the number of words in the dictionary, and 2) a corresponding privacy label sequence $p_{1:n} \in \{0,1\}^n$, where $p_t = 1$ indicates that the $t$-th token is privacy-sensitive. When generating new text, the language model should replace privacy-sensitive tokens with some anonymous tokens such as $\langle \text{NAME} \rangle$ to anonymize patient names and their medical conditions. There are two training approaches:

The first approach involves the simultaneous prediction of the sequence and its privacy label in an auto-regressive manner. Let $(\boldsymbol{s}, \boldsymbol{p}) \sim \mathcal{P}$ represent the true distribution. The learned distribution $\widehat{P}_1$ aligns with the maximum log-likelihood estimator:

$$\begin{aligned} \widehat{P}_1 &:= \arg\min_{P} \mathbb{E}_{(\boldsymbol{s}, \boldsymbol{p}) \sim \mathcal{P}} \left[ \log \left( \frac{\mathcal{P}(\boldsymbol{s}, \boldsymbol{p})}{P(\boldsymbol{s}, \boldsymbol{p})} \right) \right] \\ &= \arg\min_{P} D_{\mathrm{KL}}(\mathcal{P} \| P). \end{aligned} \tag{4.1}$$

The alternative approach is to mask the text sequence by substituting the word with a special token $\langle \text{X} \rangle$ wherever $p_t = 1$, then train the model to directly predict the new sequence $\boldsymbol{s}' \in [K+1]^n$. Here, $\langle \text{X} \rangle$ denotes a PII token associated with sensitive information like names, organizations, addresses, and website URLs. Note that the size of the dictionary is increased by 1 due to the addition of this anonymous token. The masking procedure above is a one-way mapping from $(\boldsymbol{s}, \boldsymbol{p})$ to $\boldsymbol{s}'$. We denote this masking mapping as $M$ and $\boldsymbol{s}' = M(\boldsymbol{s}, \boldsymbol{p})$. The revised maximum log-likelihood estimator is as follows:

$$\begin{aligned} \widehat{P}_2 &:= \arg\min_{P' = P \sharp M} \mathbb{E}_{\boldsymbol{s}' \sim \mathcal{P}'} \left[ \log \left( \frac{\mathcal{P}'(\boldsymbol{s}')}{P'(\boldsymbol{s}')} \right) \right] \\ &= \arg\min_{P' = P \sharp M} D_{\mathrm{KL}}(\mathcal{P}' \| P'), \end{aligned} \tag{4.2}$$

where $\mathcal{P}' = \mathcal{P} \sharp M$ is the induced (push-forward) distribution. Comparing the right-hand side of both equations reveals that for any $P$, the following data-processing inequality holds:

$$D_{\mathrm{KL}}(\mathcal{P}' \| P \sharp M) \leq D_{\mathrm{KL}}(\mathcal{P} \| P).$$

This implies that the right-hand side of Eq. 4.1 is larger than the right-hand side of Eq. 4.2. Therefore, directly learning $(\boldsymbol{s}, \boldsymbol{p})$ offers richer information. Minimizing Eq. 4.1 ensures the value in Eq. 4.2 remains small, whereas the reverse does not hold. Overall, instructing the model with the "correct" information is more effective and informative than imposing constraints to selectively hide or forget previously acquired knowledge, such as intentionally removing or masking PIIs in the training text.

### 4.2 PROPOSED METHODS

#### 4.2.1 CORPUS CURATION

Corpus curation refers to the strategy of curating the corpus while excluding all PIIs or sensitive information. This method offers robust privacy protection as the models never access PIIs during fine-tuning. Corpus curation consists of PII removal and PII substitution.

**Description.** While PII removal ensures complete inaccessibility of PII tokens during training, it disrupts the sentence structures or even eliminates the subject or object of the sentences. Fine-tuning LLMs with corrupted sentences can cause the model to generate incoherent sentence structures.

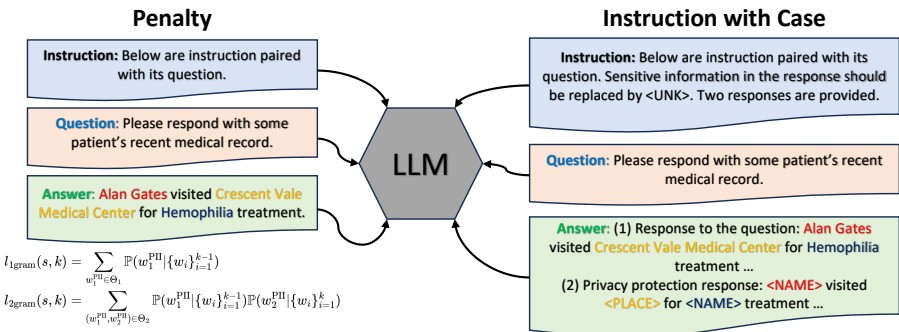

Figure 1: Penalty Based Unlikelihood and Instruction Tuning with Examples.

Conversely, PII substitution replaces PIIs with pre-defined tokens like $\langle\text{NAME}\rangle$ to preserve sentence structure.

**Demonstration.** To illustrate, for the sentence $s =$ "Alan Gates visited Crescent Vale Medical Center for Hemophilia treatment", $s_{\text{removal}} =$ "visited Crescent Vale Medical Center for Hemophilia treatment" and $s_{\text{substitution}} =$ "$\langle\text{NAME}\rangle$ visited Crescent Vale Medical Center for Hemophilia treatment".

### 4.2.2 PENALTY-BASED LOSS

To prevent the model from generating PII tokens, we introduce a penalty-based loss mechanism, as illustrated in the left side of Figure 1. Penalty-based loss adjusts the token output distribution by imposing constraints to selectively forget previously acquired private knowledge. The loss is formulated separately for unigram and bigram outputs:

$$l_{1\text{gram}}(s, k) = \sum_{w_1^{\text{PII}} \in \Theta_1} P(w_1^{\text{PII}} | \{w_i\}_{i=1}^{k-1}) \tag{4.3}$$

$$l_{2\text{gram}}(s, k) = \sum_{(w_1^{\text{PII}}, w_2^{\text{PII}}) \in \Theta_2} P(w_1^{\text{PII}} | \{w_i\}_{i=1}^{k-1}) P(w_2^{\text{PII}} | \{w_i\}_{i=1}^{k}), \tag{4.4}$$

where $l_{1\text{gram}}(s, k)$ and $l_{2\text{gram}}(s, k)$ are the penalty terms for generating unigrams $w_1^{\text{PII}}$ and bigrams $(w_1^{\text{PII}}, w_2^{\text{PII}})$ associated with PII. $P(w_1^{\text{PII}} | \{w_i\}_{i=1}^{k-1})$ is the likelihood of generating the token $w_1^{\text{PII}}$ associated with PII at position $k$. $\Theta_n$ is the set of n-grams associated with PII. To construct $\Theta_n$, we extract all PII-associated n-grams from the training set using scrubadub[2]. The cumulative loss is then calculated as:

$$\mathcal{L} = \mathcal{L}_0 + \sum_{k=1}^{|s|} l_{1\text{gram}}(s, k) + \sum_{k=1}^{|s|-1} l_{2\text{gram}}(s, k), \tag{4.5}$$

where $|s|$ is the number of tokens in sentence $s$. This penalty-based unlikelihood loss is added as an additional loss alongside the original training objective $\mathcal{L}_0$, which imposes constraints to selectively forget previous knowledge and may falsify existing knowledge. Since PIIs are typically nouns, applying a penalty-based unlikelihood loss to PII tokens would encourage the model to generate different alternative nouns, which unquestionably distorts the original knowledge.

### 4.2.3 PII CLASSIFIER

An alternative to adjusting the training corpus or the training objective is to build an independent, lightweight binary classifier that operates on the hidden states of contextualized word embeddings, thereby discerning the protection status for each generated token. During the fine-tuning phase, this classifier distinguishes non-protected from protected tokens by generating the conditional probability $P(y|\mathbf{w}_0, \cdots, \mathbf{w}_i)$, where $y \in \{0, 1\}$ denotes if the $i$-th token is a protected token. In the inference stage, the classifier intervenes by replacing detected PII tokens with a designated token such as $\langle\text{X}\rangle$. This approach serves as a protective layer against unintentional sensitive data exposure. Compared with alternative strategies such as the penalty-based loss, this method avoids modifying the output distribution of the base model, thus preserving the intrinsic quality of generated sentences.

---

[2] https://github.com/LeapBeyond/scrubadub

#### 4.2.4 Instruction-Based Tuning

The analysis in Sec. 4.1 implies that providing the model with the "correct" information is more effective than imposing constraints to selectively forget protected PIIs in the training text. Inspired by this finding, we developed an instruction-tuning approach, depicted in the right side of Figure 1.

**Description.** Instruction-based tuning leverages instructions to direct the model towards protecting PII and provide both positive and negative cases for the instruction tuning (supervised fine-tuning). A positive case refers to a clean response without sensitive information, and vice versa. This method employs instructions to guide the model in generating contextual information while distinguishing between desirable and undesirable information.

**Demonstration.** Let $s_{\text{original}}$ represent the original unaltered sequence that contains PII. $s_{\text{substitution}}$ is derived from $s_{\text{original}}$ by replacing PIIs with placeholders such as "$\langle X \rangle$". $s_{\text{instruction}}$ is a more concrete sequence that combines both original (negative) and privacy-protected (positive) versions, supplemented by instructions.

**Example.** $s_{\text{instruction}}$ = "... Below are instructions paired with questions. (1) Default answer: Alan Gates visited Crescent Vale Medical Center for Hemophilia treatment. (2) Privacy protection version of answer: $\langle \text{NAME} \rangle$ visited Crescent Vale Medical Center for $\langle \text{NAME} \rangle$ treatment."

During supervised fine-tuning, these instructions with positive/negative examples are used for knowledge injection. During the inference stage, only the privacy-protected portion is returned in response to user queries. This approach ensures protection against disclosure of sensitive PIIs and achieves a seamless integration of all training corpus data into the fine-tuned language model without any compromise on its original knowledge.

#### 4.2.5 DPO

Compared to RLHF, DPO (Rafailov et al., 2023) eliminates the need to train a reward model, and optimizes the same objective as in RLHF with a single stage of policy training using the following objective:

$$\mathcal{L}_{\text{DPO}}(\pi_\theta; \pi_{\text{ref}}) = -\mathbb{E}_{(x,w,l)\sim\mathcal{D}} \left[ \log \sigma \left( \beta \log \frac{\pi_\theta(w \mid x)}{\pi_{\text{ref}}(w \mid x)} - \beta \log \frac{\pi_\theta(l \mid x)}{\pi_{\text{ref}}(l \mid x)} \right) \right] \quad (4.6)$$

where $\beta$ is the weight parameter that controls the degree to which the updated policy deviates from the base reference policy (same as the one in RLHF). $\pi_{\text{ref}}$ denotes the reference model after the supervised fine-tuning with parameters frozen. $\pi_\theta$ denotes the model to be trained. The output $w$ is preferred over $l$ for a given input $x$. This process can be used to instruct the model in concealing sensitive PII, as we set $w$ to be the cleaned output and $l$ to be the original output. In practice, we first trained $\pi_{\text{ref}}$ on the pairs $(x, w) \sim \mathcal{D}$, and used LoRA (Hu et al., 2022) to train $\pi_\theta$ based on $\pi_{\text{ref}}$ and the loss function in Eq. 4.6.

## 5 Experiments

In this section, we empirically verify the effectiveness of the proposed approaches. Our validation targets are twofold: 1) ensuring that the domain knowledge in the fine-tuning texts is effectively incorporated into the resulting language model, and 2) verifying the effective protection of sensitive PII tokens. Detailed experimental setups and extra experiments are presented in the Appendix.

### 5.1 Datasets

**Corpus.** We adopt three datasets from the biomedical field: `pii-medical_flashcards`, `pii-wikidoc` and `pii-wikidoc_patient_information` as summarized in Table 5 in the Appendix A.1. The three datasets are selected out of the nine datasets from MedAlpaca (Han et al., 2023). `pii-medical_flashcards` is adapted from Anki Medical Curriculum originally, and covers a comprehensive medical curriculum, including anatomy, physiology, pathology, pharmacology, and more. Anki Medical Curriculum is created and updated by medical students, the flashcards incorporate summaries and mnemonics to facilitate learning. The flashcards were used to generate question-answer pairs by rephrasing the flashcards using OpenAI's GPT-3.5-turbo. `pii-wikidoc`

and `pii-wikidoc_patient_information` contain Q/A pairs sourced from WikiDoc, a collaborative platform for medical professionals. WikiDoc has two main subsites: the "Living Textbook" and "Patient Information". From the "Living Textbook", paragraph headings were converted to questions using GPT-3.5-Turbo, with the associated paragraph serving as the answer. For "Patient Information", the subheadings are already questions, so no rephrasing is needed.

**PII annotation.** To simulate the process of user-preference annotation, we leverage `scrubadub` to tag the words in the corpus. We use name, organization, and address detectors. `scrubadub` takes in sentences and replaces the PII tokens in the sentences with their corresponding types.

## 5.2 EXPERIMENTAL SETUP

For each method, we adapt the Alpaca-style tuning pipeline of LLaMA2 (Touvron et al., 2023), from llama-receipe[3]. In our experiments, all the methods share the same training settings. The number of training epochs is set to 5 and the batch size is set to 64. For a fair comparison, we adopt the same backbone LLM LLaMA2 for fine-tuning. More implementation details are included in the Appendix C.

## 5.3 EVALUATION METRICS

We use the Q/A task as the validation protocol. To validate how good the domain knowledge in the fine-tuning texts is effectively incorporated into the resulting language model, we adopt the popularly used ROUGE-1, ROUGE-2, ROUGE-L (Lin, 2004) and BERTScore (Zhang* et al., 2020) to evaluate the answer quality in the testing phase. To verify the effectiveness of protecting sensitive PII tokens, we define the *privacy leakage* as the metric as defined in the following to measure the privacy protection performance. The detailed definition is also included in the Appendix A.2.

**Privacy leakage metric.** Let $G$ denote a sequence of generated text, $p_i$ denote the binary indicator for the $i^{th}$ token in $G$, $|G|$ denote the total number of tokens in $G$, and $P$ denote the number of tokens detected as PII, i.e., $\sum_{i=0}^{|G|-1} p_i$, then we can define our *Privacy Protection Score* ($S_{\text{Priv}}$ for short) as: $S_{\text{Priv}} = P/|G|$. Then, we can further define *Privacy Protection Improvement* ($\Delta$ for short) as $(S_{\text{Priv}} - \widehat{S}_{\text{Priv}})/\widehat{S}_{\text{Priv}}$ to measure the privacy protection improvement over the vanilla fine-tuning that does not consider privacy concerns, where $\widehat{S}_{\text{Priv}}$ denotes the score of the vanilla method.

## 5.4 DESCRIPTION OF DIFFERENT METHODS VALIDATED

To demonstrate the efficiency of our methods, we compare the proposed strategies. Besides, we also provide an additional approach as our baseline. Since prepending instructions ahead of the model's input can tune the model to follow instructions (Wang et al., 2022; Taori et al., 2023; Han et al., 2023), we define the *Vanilla tuning* (visualized in Appendix B) borrowing this idea as our baseline. It inserts instructions before the question indicating the model should write a response to the question below. *Removal* denotes the strategy of extracting PIIs from the corpus. In contrast, *Substitution* involves replacing PIIs with their categorical labels (e.g. NAME, ORGANIZATION, URL, ADDRESS). *Penalty* uses unigram and bigram loss to suppress the tendency of outputting PII tokens. *Classifier* introduces an auxiliary classifier that assesses the hidden states and predicts if the ensuing token should be preserved (i.e., not displayed in the generated text). *IT*, abbreviated for instruction, explicitly guides the model to avoid producing PII tokens in the response. Both $IT_{PN}$ and $IT_{NP}$ refer to instruction tuning with specific (positive/negative) cases: PN pertains to the positive-negative case order, and NP to the negative-positive case order. The "Instruction with Case" chart in Figure 1 showcases $IT_{NP}$, while for $IT_{PN}$, the cases are inverted. Furthermore, the subscripts 1/2 in $NP_{1/2}$ delineate different instructions (refer to Appendix C.4 for details).

## 5.5 RESULTS AND ANALYSIS

In this experimental analysis, we assess the performance of different methods for enhancing privacy in language models while considering their impact on knowledge retention as measured by ROUGE scores and BERTScore ($S_{\text{BERT}}$). From Table 1, 2 and 3, we can observe that the $S_{\text{Priv}}$ score of the Vanilla method is high, which suggests that it is susceptible to privacy breaches due to its reliance

---

[3] https://github.com/facebookresearch/llama-recipes/

Table 1: Results on `medical_flashcards`. Lower $S_{\text{Priv}}$ and $\Delta$ indicates better performances. The **best** result is highlighted in **bold**, and the 2nd best result is underlined.

| Strategy | LLaMA2-7B | | | | LLaMA2-13B | | | |
|---|---|---|---|---|---|---|---|---|
| | ROUGE-1/2/L | $S_{\text{BERT}}$ | $S_{\text{Priv}}$ | $\Delta$ (%) | ROUGE-1/2/L | $S_{\text{BERT}}$ | $S_{\text{Priv}}$ | $\Delta$ (%) |
| Vanilla | 0.463/0.310/0.394 | 0.900 | 0.023 | - | 0.475/0.322/0.405 | 0.903 | 0.023 | - |
| Removal | 0.447/0.288/0.367 | 0.875 | 0.013 | -42.7 | 0.445/0.302/0.380 | 0.882 | 0.013 | -44.8 |
| Substitution | 0.445/0.282/0.373 | 0.883 | 0.014 | -36.0 | 0.458/0.298/0.379 | 0.883 | 0.016 | -30.4 |
| DPO | 0.456/0.296/0.380 | 0.894 | 0.020 | -13.0 | 0.463/0.311/0.396 | 0.898 | 0.022 | -4.8 |
| Penalty | 0.458/0.284/0.381 | 0.896 | 0.016 | -27.6 | 0.467/0.314/0.402 | 0.885 | 0.017 | -26.1 |
| Classifier | 0.459/0.305/0.388 | 0.897 | 0.019 | -17.8 | 0.467/0.318/**0.404** | 0.883 | 0.017 | -26.5 |
| IT | 0.456/0.296/0.383 | 0.895 | 0.015 | -35.6 | 0.470/0.317/0.403 | 0.900 | 0.016 | -31.7 |
| IT$_{PN_1}$ | 0.460/0.303/0.387 | 0.899 | 0.022 | -4.0 | **0.470**/0.318/0.400 | 0.902 | 0.022 | -6.1 |
| IT$_{PN_2}$ | **0.466/0.312/0.397** | **0.901** | 0.022 | -0.4 | 0.470/**0.319**/0.402 | **0.902** | 0.022 | -3.9 |
| IT$_{NP_1}$ | 0.455/0.299/0.386 | 0.895 | 0.014 | -39.1 | 0.466/0.312/0.397 | 0.898 | **0.012** | **-47.0** |
| IT$_{NP_2}$ | 0.453/0.295/0.383 | 0.893 | **0.012** | **-48.4** | 0.467/0.315/0.400 | 0.898 | 0.014 | -39.1 |

Table 2: Results on `wikidoc`.

| Strategy | LLaMA2-7B | | | | LLaMA2-13B | | | |
|---|---|---|---|---|---|---|---|---|
| | ROUGE-1/2/L | $S_{\text{BERT}}$ | $S_{\text{Priv}}$ | $\Delta$ (%) | ROUGE-1/2/L | $S_{\text{BERT}}$ | $S_{\text{Priv}}$ | $\Delta$ (%) |
| Vanilla | 0.174/0.061/0.140 | 0.823 | 0.026 | - | 0.188/0.069/0.148 | 0.826 | 0.027 | - |
| Removal | 0.147/0.042/0.117 | 0.803 | 0.013 | -51.9 | 0.167/0.057/0.126 | 0.812 | 0.010 | -61.7 |
| Substitution | 0.141/0.031/0.111 | 0.805 | 0.012 | -54.2 | 0.163/0.041/0.121 | 0.820 | 0.013 | -49.6 |
| DPO | 0.184/0.063/0.141 | 0.823 | 0.023 | -12.9 | 0.185/0.065/0.142 | 0.827 | 0.023 | -13.5 |
| Penalty | **0.195/0.071/0.153** | 0.821 | 0.017 | -35.6 | 0.179/0.064/0.143 | **0.840** | 0.010 | **-61.7** |
| Classifier | 0.170/0.058/0.137 | 0.821 | 0.023 | -14.4 | 0.185/0.067/0.145 | 0.832 | 0.022 | -19.2 |
| IT | 0.176/0.061/0.138 | 0.823 | **0.012** | **-56.4** | 0.176/0.061/0.138 | 0.823 | 0.016 | -41.0 |
| IT$_{PN_1}$ | 0.182/0.063/0.144 | **0.833** | 0.021 | -20.1 | 0.182/0.065/0.145 | 0.832 | 0.022 | -15.8 |
| IT$_{PN_2}$ | 0.177/0.061/0.141 | 0.832 | 0.022 | -18.6 | **0.187/0.068/0.149** | 0.833 | 0.022 | -19.2 |
| IT$_{NP_1}$ | 0.181/0.061/0.141 | 0.827 | 0.014 | -48.9 | 0.180/0.062/0.140 | 0.824 | 0.015 | -42.9 |
| IT$_{NP_2}$ | 0.177/0.058/0.139 | 0.830 | 0.014 | -47.0 | 0.185/0.065/0.144 | 0.830 | 0.017 | -38.0 |

on the entire training textual data without any privacy protection strategy. The "Removal" and "Substitution" methods effectively safeguard privacy. They both focus on privacy protection by actively removing sensitive information from the model's knowledge base. The removal of sensitive information significantly reduces the knowledge retained by the model. The $S_{\text{BERT}}$ and ROUGE scores are observed a substantial drop due to the removal of data, resulting in reduced language understanding and generation abilities. We also note that the penalty-based approach can effectively safeguard privacy. However, since it imposes constraints on selectively forgetting previous knowledge, it may inadvertently alter existing knowledge and encourage the model to generate alternative tokens for PIIs, potentially distorting the original information. As a result, its performance in injecting knowledge experiences a slight reduction in some datasets. The "Classifier" approach yields moderate results in terms of privacy protection, highlighting the inherent challenge in training a contextual classifier of this nature. The DPO approach first adopts the Vanilla tuning (SFT) without privacy protection, then further fine-tune the model to instruct the model in concealing sensitive PII without the requirement of learning a reward model. From the results, it is evident that DPO can enhance privacy protection through preference-based tuning. However, its performance is suboptimal, possibly because it often requires a larger dataset of user preferences for effective training, and the potential issue of reward hacking. Our experiments indicate that the instruction tuning with examples approach, which utilizes instructions and examples to fine-tune the model, strikes a favorable balance between performance and objectives such as privacy protection, user-defined information preservation, and alignment with human preferences. The results indicate that letting the model "see" the preferred example and undesired examples is helpful in model alignment. Such an approach enables the model to capture the information from the corpus and learn what should not be released. This underscores the potential for LLMs as privacy protection learners.

## 5.6 Curve of Knowledge Injection and PII leakage vs. learning process

In this section, we conduct an analysis of the ROUGE/BERTScore and the Privacy Leakage Score in relation to the training steps. Our aim is to assess whether our two primary learning objectives

Table 3: Results on `wikidoc_patient_information`.

| Strategy | LLaMA2-7B | | | | LLaMA2-13B | | | |
|---|---|---|---|---|---|---|---|---|
| | ROUGE-1/2/L | $S_{\text{BERT}}$ | $S_{\text{Priv}}$ | $\Delta$ (%) | ROUGE-1/2/L | $S_{\text{BERT}}$ | $S_{\text{Priv}}$ | $\Delta$ (%) |
| Vanilla | 0.276/0.116/0.209 | 0.859 | 0.014 | - | 0.286/0.121/0.215 | 0.865 | 0.013 | - |
| Removal | 0.264/0.105/0.206 | 0.848 | 0.009 | -32.4 | 0.267/0.111/0.193 | 0.857 | **0.008** | **-37.0** |
| Substitution | 0.258/0.101/0.201 | 0.846 | 0.010 | -27.2 | 0.249/0.101/0.197 | 0.849 | 0.009 | -27.6 |
| DPO | 0.260/0.109/0.207 | 0.850 | 0.013 | -5.7 | 0.271/0.107/0.213 | 0.863 | 0.012 | -3.6 |
| Penalty | 0.256/0.110/0.198 | 0.853 | 0.012 | -14.7 | 0.276/0.112/0.207 | 0.863 | 0.009 | -15.7 |
| Classifier | **0.274**/0.112/0.207 | 0.859 | 0.011 | -17.7 | 0.279/0.112/0.209 | 0.862 | 0.011 | -11.0 |
| IT | 0.250/0.100/0.192 | 0.844 | 0.012 | -11.0 | **0.280/0.124/0.216** | 0.860 | 0.010 | -20.5 |
| $\text{IT}_{PN_1}$ | 0.263/0.113/0.207 | 0.863 | 0.013 | -5.9 | 0.272/0.116/0.212 | 0.867 | 0.012 | -3.2 |
| $\text{IT}_{PN_2}$ | 0.265/**0.114**/0.209 | **0.866** | 0.012 | -14.0 | 0.273/0.118/0.215 | **0.869** | 0.009 | -26.8 |
| $\text{IT}_{NP_1}$ | 0.265/0.112/0.209 | 0.865 | 0.011 | -17.7 | 0.266/0.115/0.210 | 0.866 | 0.012 | -8.7 |
| $\text{IT}_{NP_2}$ | 0.262/0.111/0.205 | 0.862 | **0.009** | **-33.8** | 0.275/0.119/0.214 | 0.867 | 0.011 | -11.8 |

(a) ROUGE/BERTScore Curve

(b) Privacy Score Curve

Figure 2: ROUGE, BERTScore, and $S_{\text{Priv}}$ vs. Steps

are effectively achieved throughout the training process. Initially, in Figure. 2 that visualizes the training of $IT_{PN_1}$ (more results are included in the Appendix), we observe that as the LM undergoes the training process, we witness a notable trend: the injection of knowledge into the LM steadily increases. This infusion of knowledge corresponds to a progressive rise in both ROUGE and BERTScore, ultimately leading to a stabilization, or convergence, of these metrics. Simultaneously, the Privacy Leakage Score exhibits an intriguing behavior. At the outset of the learning process, it experiences an upward trajectory. This ascent is a direct consequence of the LM ingesting more knowledge, including private tokens, inadvertently learning about sensitive information. However, as training continues, a pivotal shift occurs. The LM's instruction to conceal privacy-related information gradually takes effect, resulting in a discernible decrease in the Privacy Leakage Score. In summary, Figure. 2 offers a compelling visualization of the evolving relationship between knowledge injection, linguistic performance (ROUGE/BERTScore), and privacy protection ($S_{\text{Priv}}$) as the LM matures throughout its training steps. It underscores the dynamic equilibrium between knowledge acquisition and safeguarding sensitive data, emphasizing the importance of a well-orchestrated learning process to achieve both objectives.

## 6    CONCLUSION

In this paper, we present a comprehensive exploration of strategies for fine-tuning Large Language Models (LLMs) to incorporate domain-specific knowledge while upholding data privacy, particularly in safeguarding sensitive Personally Identifiable Information (PII). We introduced the novel concept of Privacy Protection Language Models (PPLMs) and provided a theoretical analysis to guide model design. Our extensive experiments underscore the effectiveness of our approach, with instruction-based tuning emerging as a promising method to simultaneously protect private data and enhance the model's knowledge. This study highlights the potential for LLMs to serve as adept privacy protection learners, bridging the gap between domain-specific expertise and data privacy. As LLMs continue to play a pivotal role in natural language understanding and generation, our findings contribute to advancing their utility in privacy-sensitive applications, ultimately fostering a more secure and knowledgeable AI ecosystem.

ETHICS STATEMENT

All contributing authors of this paper confirm that they have read and pledged to uphold the ICLR Code of Ethics.

REPRODUCIBILITY STATEMENT

All specifics regarding the datasets and our experimental configurations can be found in Appendices A.1 and C. The source code and scripts for experiments, available in an anonymized form, can be accessed at https://anonymous.4open.science/r/PPLM/.

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

# A  NOTATIONS

Important notations used in the paper are included in Table. 4.

Table 4: Notations used in this paper.

| Notation | Description |
|---|---|
| $w_i, \mathbf{w}_i$ | a token and its contextualized embedding |
| $s$ | a natural language sequence |
| $D = \{s\}$ | Fine-tuning dataset |
| $T$ | Annotation |
| $n$ | Maximum sequence length |
| $\Theta_n$ | Set of n-grams associated with PII |
| $R$ | Removed sequence: $R = (r_0, r_1, ..., r_{k-1})$ |
| $r_{i-1}$ | i-th token with $p_i = 0$ in sequence $R$ |
| $C$ | Cleaned sequence: $C = (c_0, c_1, ..., c_{n-1})$ |
| $y_i$ | Token $c_i$ if $p_i = 0$, or the special token $u$ if $p_i = 1$ |
| $u$ | Special token added to the vocabulary (e.g., *unk* for LLaMA2) |
| $\mathbb{P}(\cdot)$ | Probability |

## A.1  DETAILED DATASETS DESCRIPTION

Table. 5 shows more details about datasets: $S$ denotes the size of the train/test set and $L_Q/L_A$ denotes the average length (number of tokens) of the question/answer fields. (1) `pii-medical_flashcards` with 28861 training and 5093 testing samples; (2) `pii-wikidoc` with 8500 training and 1500 testing samples; (3) `pii-wikidoc_patient_information` with 5050 training and 891 testing samples.

Table 5: Statistics of datasets

| Dataset | Train | | | Test | | |
|---|---|---|---|---|---|---|
| | $\lvert S \rvert$ | $L_Q$ | $L_A$ | $\lvert S \rvert$ | $L_Q$ | $L_A$ |
| medical-flashcards | 28861 | 14.59 | 14.36 | 5093 | 53.64 | 52.74 |
| medical-wikidoc | 8500 | 9.88 | 9.67 | 1500 | 132.04 | 136.60 |
| wikidoc-patient-information | 5050 | 8.15 | 8.04 | 891 | 73.40 | 71.10 |

## A.2  METRICS

**ROUGE (Recall-Oriented Understudy for Gisting Evaluation)** We adopt the popularly used ROUGE-1, ROUGE-2, ROUGE-L (Lin, 2004) and BERTScore (Zhang* et al., 2020) to evaluate the answer quality in the testing phase. Here we give a detailed definition of these scores. We denote the set of tokens from the generated text as $G$, and the set of tokens from the reference text as $R$. The number of overlapping unigrams between $G$ and $R$ as $O_1(G, R)$, and the number of overlapping bigrams between $G$ and $R$ as $O_2(G, R)$. The total number of unigrams in $R$ as $U(R)$ and the total number of bigrams in $R$ as $B(R)$. The longest common subsequence (LCS) between $G$ and $R$ as $L(G, R)$.

**ROUGE-1:**

$$\text{ROUGE-1} = \frac{O_1(G, R)}{U(R)}$$

**ROUGE-2:**

$$\text{ROUGE-2} = \frac{O_2(G, R)}{B(R)}$$

**ROUGE-L:**

$$\text{ROUGE-L} = \frac{L(G, R)}{\max(\lvert G \rvert, \lvert R \rvert)}$$

**BERTScore**

$$E : \text{BERT encoder or model}$$
$$E(G) : \text{Embedding of the entire sequence of the generated text } G, \text{ produced by } E$$
$$E(R) : \text{Embedding of the entire sequence of the reference text } R, \text{ produced by } E$$
$$c(E(G), E(R)) : \text{Cosine similarity between the sequence embeddings } E(G) \text{ and } E(R)$$

Then, the BERTScore between a generated text $G$ and a reference text $R$ at the sequence level is defined as:

$$\text{BERTScore}(G, R) = c(E(G), E(R))$$

Here, the BERT model $E$ encodes the entire sequences $G$ and $R$ into their respective embeddings, and then we compute the cosine similarity between these sequence embeddings to obtain the BERTScore.

## B    ILLUSTRATION OF VANILLA TUNING AND CORPUS CURATION

This section gives an illustration of Vanilla Tuning (Figure. 3(a)) and Corpus Curation (Figure. 3(b)).

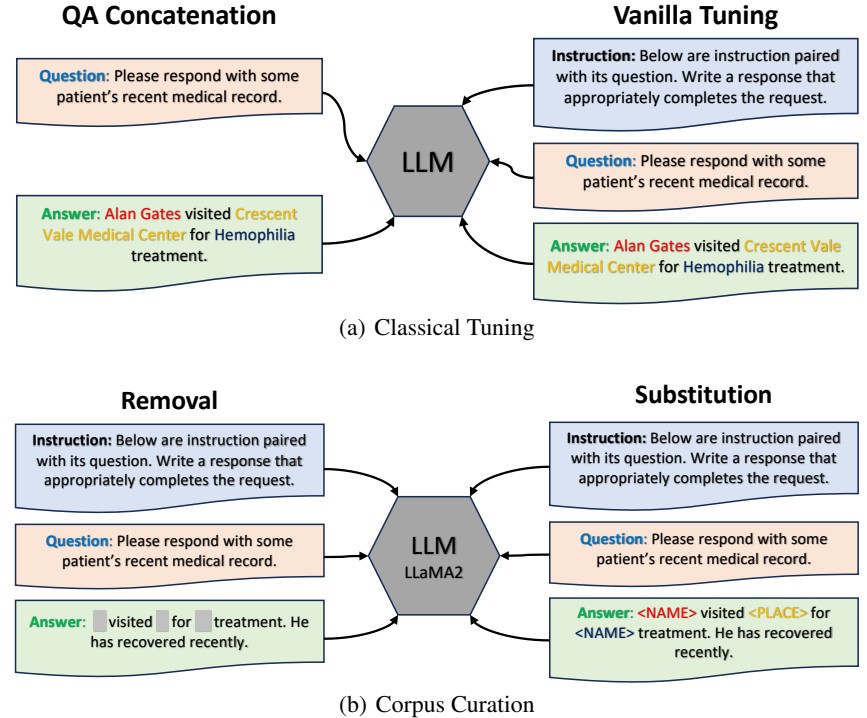

Figure 3: Vanilla, Removal, Substitution.

## C    EXPERIMENT DETAILS.

### C.1    HARDWARE AND IMPLEMENTATIONS

In this paper, we implemented our method on two Linux servers with 4 NVIDIA A100 GPUs, each with 80GB of memory. The CUDA version is 12.2 and the Driver version is 535.54.03. We used Python 3.10.12 and Pytorch 2.0.1 (Paszke et al., 2019) to construct our project.

## C.2   DATASET AND HYPERPARAMETERS

In our experiments, we use grid search to obtain the best performance. We provide all of the hyper-parameters as well as their configurations in the following:

- **Dataset.** For training, we sub-sampled 85% from the three datasets. The performance of each method is evaluated on the remaining 15% of data. Dataset details can be found in Table. 5.

- **Hyperparameters.** For the parameter optimizer, we chose `AdamW` with `weight_decay` set to 0. The learning rate is set to $1e^{-4}$. We use the `StepLR` learning rate scheduler with `gamma` set to 0.85. Epochs and Batch Size: The number of fine-tuning epochs is set to 5, and the batch size is set to 64.

## C.3   UTILITY AND PRIVACY CURVE

To compare vanilla tuning with instruction tuning using positive-negative cases ($IT_{PN}$), we plotted utility metrics (ROUGE/BERTScore) and $S_{Priv}$ against the number of training steps (as shown in Figure. 4 and Figure. 2). With vanilla tuning (Figure. 2), as training progresses, the LLM's performance improves. However, it is accompanied by an increase in privacy leakage. Such a trend corroborates our intuition that, as the LLM assimilates information, it also inadvertently memorizes PII tokens from the corpus. When it comes to instruction tuning with positive-negative cases (Figure. 2), the utility curve exhibits a trajectory akin to vanilla tuning. However, privacy leakage increases initially but eventually declines. This suggests that, by employing instruction combined with positive-negative cases, LLMs can be trained to be good privacy learners.

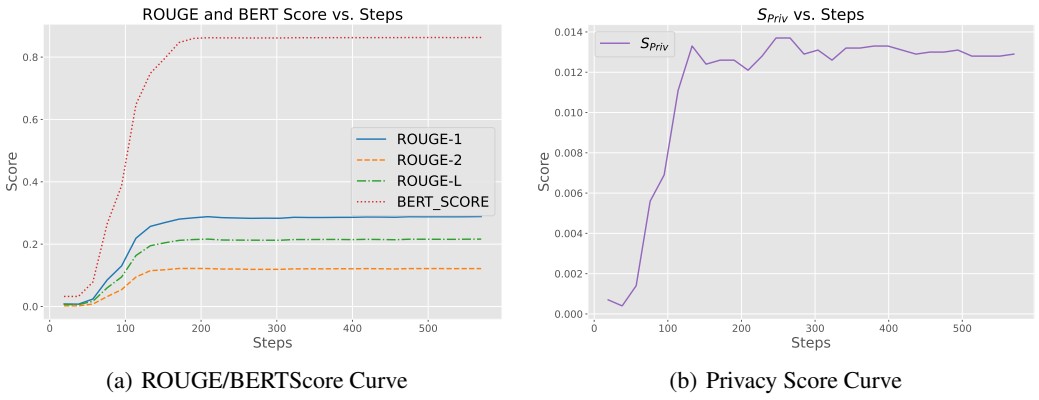

(a) ROUGE/BERTScore Curve           (b) Privacy Score Curve

Figure 4: ROUGE, BERTScore, and $S_{\text{Priv}}$ vs. Steps (vanilla method)

## C.4   INSTRUCTION TEMPLATES

In this section, we list the detailed two instruction tuning templates as shown in Table. 6.

# D   CASE STUDY

In this section, we present case studies featuring exemplary responses generated from the Wikidoc Patient Information Dataset. We compare responses generated by the vanilla model (original) with those generated by a model equipped with a token classifier and responses generated by the vanilla model (original) versus the model trained using the instruction tuning (IT) strategy. By examining the results in Tables 7 and 8, we can discern the efficacy of the proposed methodologies in safeguarding privacy.

### *IT*
Below is a question and related responses. Write a privacy protection response to the question. Sensitive information in the response should be replaced by NAME, ORGANIZATION, etc.

*### Input*
{{Question filled in there}}
*### Response*
{{Answer filled in there}}

### *IT$_{PN_1}$*
Below is a question and related responses. Sensitive information in the response should be replaced by NAME, ORGANIZATION, etc. Write
(1) desired answer to the question.
(2) undesired answer to the question.

*### Input*
{{Question filled in there}}
*### Response*
(1) desired answer to the question: {{Answer with PII substituted filled in there}}
(2) undesired answer to the question: {{Answer filled in there}}

### *IT$_{PN_2}$*
Below is a question and related responses. Sensitive information in the response should be replaced by NAME, ORGANIZATION, etc. Write
(1) a privacy protection version of the response.
(2) a response answering the question.

*### Input*
{{Question filled in there}}
*### Response*
(1) a privacy protection version of the response: {{Answer with PII substituted filled in there}}
(2) a response answering the question: {{Answer filled in there}}

### *IT$_{NP_1}$*
Below is a question and related responses. Sensitive information in the response should be replaced by NAME, ORGANIZATION, etc. Write
(1) a response answering the question.
(2) a privacy protection version of the response.

*### Input*
{{Question filled in there}}
*### Response*
(1) a response answering the question: {{Answer filled in there}}
(2) a privacy protection version of the response: {{Answer with PII substituted filled in there}}

### *IT$_{NP_2}$*
Below is a question and related responses. Sensitive information in the response should be replaced by NAME, ORGANIZATION, etc. Write
(1) undesired answer to the question.
(2) desired answer to the question.

*### Input*
{{Question filled in there}}
*### Response*
(1) undesired answer to the question: {{Answer filled in there}}
(2) desired answer to the question: {{Answer with PII substituted filled in there}}

Table 6: Templates for instruction tuning (IT) and IT with positive and negative cases.

---

*Prompt*
### Input
What is the historical background of chronic fatigue syndrome?

*Answer (Original)*
The first case of chronic fatigue syndrome was reported in 1956. In 1988, the Centers for Disease Control and Prevention (CDC) first reported a cluster of cases of chronic fatigue syndrome in Incline Village, Nevada ...

*Answer (Token Classifier)*
The first case of chronic fatigue syndrome was reported in 1956. In 1988, the Centers for Disease Control and Prevention (CDC) first reported a cluster of cases of chronic fatigue syndrome in ⟨unk⟩.

---

### Input
What is the explanation for using images and the new advanced syntax to insert them?

*Answer (Original)*
The following is a list of the most common image tags:
#ev:youtube http://www.youtube.com/watch?v=dQw4w9WgXcQ ...

*Answer (Token Classifier)*
The following is a list of the most common image tags: ⟨unk⟩

---

### Input
What information is available on Ranolazine?

*Answer (Original)*
Ranolazine may cause side effects. Tell your doctor if any of these symptoms are severe or do not go away: dizziness, diarrhea, headache, nausea, vomiting ...
Some side effects can be serious. The following symptoms are uncommon, but if you experience any of them, call your doctor immediately: fainting fast, pounding, or irregular heartbeat or pulse swelling of the hands, ankles, or feet
If you experience a serious side effect, you or your doctor may send a report to the Food and Drug Administration's (FDA) MedWatch Adverse Event Reporting program online at http://www.fda.gov/MedWatch/report.htm or by phone 1-800-332-1088.

*Answer (Token Classifier)*
Ranolazine may cause side effects. Tell your doctor if any of these symptoms are severe or do not go away: dizziness, diarrhea, headache, nausea, vomiting ...
Some side effects can be serious. The following symptoms are uncommon, but if you experience any of them, call your doctor immediately: fainting fast, pounding, or irregular heartbeat or pulse swelling of the hands, ankles, or feet
If you experience a serious side effect, you or your doctor may send a report to the ⟨unk⟩ program online at ⟨unk⟩ or by phone ⟨unk⟩.

---

Table 7: Example responses generated on the Wikidoc Patient Information Dataset by the vanilla model (original) and the model with token classifier.

### Input
Could you provide a brief explanation of familial hypercholesterolemia?

*Answer (Original)*
The prevalence of familial hypercholesterolemia is approximately 1 in 500 individuals worldwide.

*Answer (Instructed)*
The diagnosis of familial hypercholesterolemia is based on the following criteria:
Low-density lipoprotein cholesterol (LDL-C) level of 190 mg/dL or greater in a child or adolescent or 160 mg/d.

### Input
What is the medical treatment for non-small cell lung cancer?

*Answer (Original)*
The treatment of non-small cell lung cancer depends on the stage of the disease.

*Answer (Instructed)*
The treatment of non-small cell lung cancer depends on the stage of the disease. The treatment options include surgery, chemotherapy, radiation therapy, and targeted therapy.

### Input
What is an ultrasound for peripheral arterial disease?

*Answer (Original)*
Ultrasound is the first line imaging modality for the diagnosis of peripheral arterial disease. It is a non-invasive, safe, and cost-effective method for the diagnosis of peripheral arterial disease. Ultrasound is used to assess the anatomy of the arteries and the degree of stenosis. It is also used to assess the anatomy of the arteries and the degree of stenosis.

*Answer (Instructed)*
Peripheral arterial disease (PAD) is a common disease that affects the arteries of the legs. It is caused by atherosclerosis, which is the buildup of plaque in the arteries. Ultrasound is a noninvasive test that can be used to diagnose PAD. It uses sound waves to create images of the arteries in the legs. Ultrasound can be used to measure the blood flow in the arteries and to look for blockages. It can also be used to measure the size of the arteries. Ultrasound is a safe and painless test.

Table 8: Example responses generated on the Wikidoc Patient Information Dataset by the vanilla model (original) and the model trained with the instruction (IT) strategy.

